# Impacts of Gut Microbiota on the Immune System and Fecal Microbiota Transplantation as a Re-Emerging Therapy for Autoimmune Diseases

**DOI:** 10.3390/antibiotics11081093

**Published:** 2022-08-12

**Authors:** Ashenafi Feyisa Beyi, Michael Wannemuehler, Paul J. Plummer

**Affiliations:** 1Department of Veterinary Microbiology and Preventative Medicine, College of Veterinary Medicine, Iowa State University, Ames, IA 50011, USA; 2National Institute of Antimicrobial Resistance Research and Education, Iowa State University, Ames, IA 50011, USA; 3Nanovaccine Institute, Iowa State University, Ames, IA 50011, USA; 4Department of Veterinary Diagnostic and Production Animal Production, College of Veterinary Medicine, Iowa State University, Ames, IA 50011, USA

**Keywords:** antimicrobial resistance, autoimmune diseases, fecal microbiota transplantation, hygiene theory, gut microbiota, immune system, one health

## Abstract

The enormous and diverse population of microorganisms residing in the digestive tracts of humans and animals influence the development, regulation, and function of the immune system. Recently, the understanding of the association between autoimmune diseases and gut microbiota has been improved due to the innovation of high-throughput sequencing technologies with high resolutions. Several studies have reported perturbation of gut microbiota as one of the factors playing a role in the pathogenesis of many diseases, such as inflammatory bowel disease, recurrent diarrhea due to *Clostridioides difficile* infections. Restoration of healthy gut microbiota by transferring fecal material from a healthy donor to a sick recipient, called fecal microbiota transplantation (FMT), has resolved or improved symptoms of autoimmune diseases. This (re)emerging therapy was approved for the treatment of drug-resistant recurrent *C. difficile* infections in 2013 by the U.S. Food and Drug Administration. Numerous human and animal studies have demonstrated FMT has the potential as the next generation therapy to control autoimmune and other health problems. Alas, this new therapeutic method has limitations, including the risk of transferring antibiotic-resistant pathogens or transmission of genes from donors to recipients and/or exacerbating the conditions in some patients. Therefore, continued research is needed to elucidate the mechanisms by which gut microbiota is involved in the pathogenesis of autoimmune diseases and to improve the efficacy and optimize the preparation of FMT for different disease conditions, and to tailor FMT to meet the needs in both humans and animals. The prospect of FMT therapy includes shifting from the current practice of using the whole fecal materials to the more aesthetic transfer of selective microbial consortia assembled in vitro or using their metabolic products.

## 1. Introduction

The resident gut microbiota is composed of a complex consortium of microorganisms, including bacteria, archaea, fungi, protozoans, and viruses, which reside along the alimentary tracts of humans and animals. A newborn acquires a seed microbiota mainly from the vagina of the mother during birth [1]. Following that, the gut microbiota increases in diversity and reaches adult-like compositions and structures at the age of three to five years in humans [2]. Studies show that gut microbiota is crucial for human and animal health, food digestion, production of some useful metabolites, colonization resistance toward pathogenic bacteria, and the development of innate and adaptive immune systems [1,3]. However, several factors, such as diet, antibiotics, lifestyle, surgical interventions, and bacterial or viral infections, alter the microbial composition and, subsequently, may have detrimental consequences on health.

Gut microbiota has a significant association with autoimmune diseases, including inflammatory bowel disease (IBD), asthma, multiple sclerosis, and type 1 diabetes [4,5,6], and other health problems, such as cancer, mental health, and cardiovascular, respiratory, and metabolic diseases, which were not expected to have associations with the microbiota. Furthermore, disruption of gut microbial diversity and composition has been shown to lead to the occurrence of several health problems including autoimmune diseases [7,8,9,10]. Unfortunately, for several of autoimmune diseases, there are no effective or lasting medications. Nevertheless, fecal microbiota transplantation (FMT), where gut microbiota from a healthy person is administered to a sick person to re-establish healthy normal microbiota after infection by pathogens, such as *Clostridioides difficile,* has recently gained popularity in clinical medicine. The newly acquired breakthrough in effectively treating *C. difficile* infections using FMT has inspired scientists to develop a treatment for autoimmune diseases and multidrug-resistant infections using this (re)emerging therapy.

Recently, gut microbiota and FMT have received attention from the public, and the “do-it-yourself” FMT movement has been noticed. According to an online survey conducted from January 2018 to February 2019, 82% of 84 respondents involved in this movement have claimed improvement in their conditions. The two main conditions for which the therapy was sought were IBD and irritable bowel syndrome (IBS) [11]. However, it should be noted that such treatments could be dangerous unless the procedures are approved by FDA and conducted in licensed medical centers. On the other hand, FMT positively affects decolonization of antimicrobial-resistant bacterial infections (see a review by Davido et al. [12]). FMT was shown to effectively induce decolonization of specific pathogens in 66.7% (102/153) of patients who received this therapy between 2000 and 2020 [13].

The transfer of rumen cud, called transfaunation, from healthy to sick animals was practiced long before the rumen microorganisms were understood [14]. Rumen transfaunation is indicated for indigestion and improves feed conversion efficiency in cattle [15,16]. FMT has been attempted in dogs and cats to control autoimmune diseases [17,18], in chickens to mitigate the shedding and spread of foodborne pathogens [19], and in piglets to increase their resistance to infections [20,21]. This review highlights the effects of gut microbiota on the development of the immune system and the potential use of FMT for the treatment of autoimmune diseases and multidrug-resistant infections in humans and animals. Figure 1 depicts the interrelationships between gut microbiota, its beneficial roles, factors that affect microbial diversity and composition, and the benefits and challenges of FMT.

## 2. Gut Microbiota, Immune Development, and Fecal Microbiota Transplantation

### 2.1. Overview of Gut Microbiota in Humans

Gut microbiota impacts functional and immunological development in humans [22]. Their beneficial roles include digestion of cellulose, production of short-chain fatty acids (SCFAs), resistance to colonization by pathogenic organisms, and maintaining immune homeostasis [23,24,25]. Gut microbiota, represented by fecal microbiota in most studies for simplicity of sample collections, consists mainly of four phyla of bacteria: Firmicutes, Bacteroidetes, Proteobacteria, and Actinobacteria. According to one study, these four phyla comprise 93.5% of 2172 species that were classified into 12 different phyla in humans [26]. Firmicutes and Bacteroidetes account for over 80% of the microbiota. Firmicutes comprises mainly Gram-positive bacteria but also a low number of Gram-negative species (e.g., *Veillonella* species). Bacteroidetes consists of Gram-negative bacteria, primarily represented by the *Bacteroides* genus and other frequently detected genera, including *Alistipes, Parabacteroides, Prevotella, Tannerella,* in the human gut [27,28,29].

#### 2.1.1. Establishment of Gut Microbiota after Birth

Colonization and the establishment of gut microbiota in humans occur gradually during early life. Fetuses acquire their initial microbiota in utero, which is followed by colonization after delivery from the environment during childbirth, primarily from the vagina [30,31]. Moles and colleagues [30] showed the difference in bacterial types between the meconium of prenatal infants and feces from their first three weeks of life. They demonstrated that postnatal bacterial exposures have primary roles in the establishment of gut microbiota in infants. The gut microbial composition of infants is stable compared to that of adults until the transition from breastfeeding to solid food, which induces remarkable shifts [32,33]. After the achievement of adult-like gut microbial composition at the age of three to five years [2], the microbial diversity shows little variation throughout adult life unless there are disruptions affecting change, such as diet, antibiotics, infections, cancer, or surgery [34].

#### 2.1.2. Factors Affecting Diversities and Compositions of Gut Microbiota

The perturbation of gut microbiota may lead to a decrease in microbial diversity and a shift of composition and cause an abnormal condition called dysbiosis, which is associated with several disease conditions [35]. As mentioned above, various factors cause the alteration of microbial communities [9,27,36]. There is even a marked difference in microbial compositions between children born by cesarean section and those born naturally, where the former has been shown to have a negative side effect on microbial diversity [6,37]. Nevertheless, diets and antibiotics are the primary causes of dysbiosis in humans; thus, a brief account of their impacts on gut microbiota is presented below.

i.Diets

The effects of diet on gut microbiota are not uniformly agreed upon across published studies; however, a large body of scientific literature has documented both negative and positive impacts of various foods on microbiota [24,35,38,39,40]. Consumption of fiber-rich or plant-originated foods, such as vegetables and fruits, as compared to meats and dairy products, has different effects on the compositions of microbial communities [38,39].

In a study that compared the microbiota of children in Europe and rural Africa (i.e., Western diets and agrarian diets, respectively), distinct patterns in microbial diversity and composition were observed [24]. In the African children, genera associated with the digestion of cellulose, such as *Prevotella* species and *Xylanibacter* species, had significantly higher abundance than in the Europeans. Similarly, a higher amount of SCFAs was produced by the gut microbiota in the African children [24]. Short-chain fatty acids, such as butyrate, acetate, and propionate, have been shown to be beneficial to humans because they have anti-inflammatory effects and enhance intestinal barrier functions by increasing mucus production and enterocyte tight junctions [41,42]. *Enterobacteriaceae*, such as *Shigella* and *Escherichia*, that have unfavorable health impacts were significantly lower in the African children when compared to European children, suggesting that the establishment of such pathogenic bacteria was prevented by the SCFA-producing bacteria that were abundant in the gut of the Africans [24]. This study demonstrated that diet plays a critical role in shaping the diversity and composition of gut microbiota.

In agreement with these findings, another study reported the enrichment of beneficial gut bacteria and increased production of SCFAs related to consumption of Mediterranean diet (i.e., high-level of cereals, fruit, vegetables, and legumes) compared to a typical Western diet (i.e., animal-fat and protein-based diets) [40]. These and several other studies have provided strong evidence that consumption of processed and high-calorie carbohydrates and meats is associated with the depletion of beneficial gut bacteria from the intestinal microbiota [43,44,45].

ii.Antibiotics

Similar to diets, the indiscriminate action of antibiotics on both beneficial (i.e., mutualists) and noxious bacteria leads to remarkable changes in gut microbiota [46]. Antibiotics cause the shifting of the microbiota due to antibiotic sensitivity and inability of affected organisms to recover (i.e., lack of resilience) [47,48]. In addition, the selection of resistant bacteria already presents in the gut or acquisition of exogenous resistant bacteria, owing to the weakening of colonization resistance to external pathogenic bacteria, can perturb gut microbiota as a consequence of antibiotic administration [36,49,50]. An observational study in human patients showed that ciprofloxacin treatment affected the abundance of gut microbiota for more than six months after the termination of the medication, resulting in a substantial reduction in bacterial diversity [25]. Similarly, treatment with broad-spectrum antibiotics such as fluoroquinolones and β-lactams reduced the diversity of gut microbiota by 25%. It was shown that broad-spectrum antibiotic use caused an imbalance of the microbial composition; for instance, the most predominant genus shifted from *Faecalibacterium* to *Bacteroides* [27], which would have metabolic effects, such as less SCFAs.

Antibiotics have both short- and long-term effects on gut microbiota [51,52]. Haak and colleagues [52] reported that the return of microbiota to the pretreatment composition in patients on a combination of antibiotic therapy had taken 8 to 31 months. A mouse model study shows that prenatal exposure of mice to an oral antibiotic in late gestation influences the microbial composition of the neonates [53]. To this end, the route of antibiotic administration has differential impacts on gut microbiota, with the oral route inducing substantial changes in microbial diversities compared to parenteral routes of administration [54]. The oral route of antibiotic delivery exposes the gut microbiota to a higher concentration of drugs before they undergo catabolism, in contrast to parenteral drug administration routes. However, studies have demonstrated that parenteral delivery of antibiotics also has an impact on the gastrointestinal microbiota [55,56]. In our recent animal study, we observed that the concentrations of subcutaneously administered fluoroquinolones found in feces were several folds higher than that of plasma [57].

### 2.2. Effects of Gut Microbiota on the Immune System in Humans

#### 2.2.1. Role of Gut Microbiota in the Development of the Immune System

Colonization of the gut by microbiota starts before birth in infants, as evidenced by the detection of bacteria in the meconium of preterm neonates [30]. The maturation of the immune system coincides with the “evolvement” of gut microbiota, which happens in the first three to five years of age [32]. The gut bacteria influence the development of both innate and adaptive immunity [58,59,60]. Moreover, the effects of the microbiota extend to the peripheral sites as well as central lymphoid tissues, including bone marrow, where hematopoiesis can be affected [61,62,63,64].

The role of gut microbiota in the development and maturation of immunity has been well demonstrated following the colonization of germ-free (GF) mice. Proportion and differentiation of myeloid progenitor cells were low in GF mice, which led to higher bacterial load and acute death following a challenge with *Listeria monocytogenes* compared to the conventionally reared (CONV-R) mice [61]. However, the mechanisms by which the gut microbiota control the immune development in distal sites, such as in the bone marrow, are not well understood. In the GF mice, the defects of myelopoiesis were restored, and mice acquired resistance to *L. monocytogenes* after colonization by complex microbiota [61]. The same study revealed the effects of gut microbiota on innate immunity through directing hematopoiesis, polymorphonuclear cells in particular.

There is an interaction between gut microbiota and antimicrobial peptides [65]. Studies show that the production of antimicrobial peptides was substantially low in GF mice, and lymphoid tissues in the spleen, thymus, and lymph nodes had defects. At the same time, macrophage phagocytosis was only minimally affected compared to the CONV-R mice [34,58,59]. The gut microbiota affects specific immune cells and mediators as well. For instance, in one study, it was shown that GF mice lack IgA-secreting plasma cells, which expanded to the levels similar to that of the CONV-R mice after repopulating the gut with specific bacteria [60].

Specific bacteria have also been found to have a modulatory effect on T cell development. The polysaccharide A (PSA) produced by *Bacteroides fragilis* induced balanced numbers of T helper 1 (Th1) and T helper 2 (Th2) cells, which is mediated by TLR2/TLR1 heterodimer on lamina propria dendritic cells [3]. When these bacteria colonized GF mice, the immunomodulatory activities of PSA corrected systemic T cell deficiencies and the imbalance of Th1 and Th2 cells. As detailed by the same authors, this symbiotic factor binds to the TLR2/1 heterodimer and activates dendritic cells, which that direct CD4^+^ T cells to assume a T_reg_ cell phenotype and to secrete more anti-inflammatory cytokines [66]. These immunomodulatory activities were induced by other gut bacteria as well, such as *Lactococcus* and *Bifidobacterium* species [3].

Furthermore, the effects of gut microbiota extend to regulation of the T cell receptor (TCR) recruitment and proliferation [34]. In GF mice, the numbers of αβTCR^+^ and γδTCR^+^ T cells in intestinal intraepithelial lymphocyte population were lower than that of CONV-R mice [67]. It has been also demonstrated that colonization of GF mice with bacteria gradually increased the numbers of cells expressing either receptor; within a month, the ex-GF mice reached similar levels of T cells comparable with that of the CONV-R mice [34,67]. The influence of gut microbiota on adaptive immunity extends to other cells besides T cells; they influence the development of B and invariant natural killer T (iNKT) cell populations, as demonstrated in the GF mice [61].

Aside from direct effects, gut microbes also have an indirect impact on immunity through their fermentation products such as SCFAs. These SCFAs can bind to host cell receptors or induce epigenetic changes in host DNA, which result in activation or repression of the host immune genes [1]. Additionally, a host-specific microbiota is required for the development of a healthy immune system [68,69]. GF mice colonized by human microbiota had low levels of CD4^+^ and CD8^+^ T cells, few proliferating T cells, few dendritic cells, and low antimicrobial peptide expression in the intestine. In contrast, colonization of GF mice with mouse-derived segmented filamentous bacteria (SFB) restored Th17 cell numbers to that observed in CONV-R mice, suggesting that mouse-specific organisms may be necessary to attain full immune maturation in mice [68].

#### 2.2.2. Effects of Perturbation of Gut Microbiota on the Immune System

Exposure to complex microbiota during early life, when an individual’s immunity is developing and expanding, has a positive association with immune tolerance later in life [34]. Early sensitization of the immune system by exposure to microbes reduces allergic reactions in adult life [32]. Moreover, literature shows that the presence of microbes prior to weaning is more critical than colonizing after weaning [70,71]. As discussed above, one of the factors that cause perturbation of gut microbiota is exposure to antibiotics. The effect of antibiotics on autoimmune diseases such as IBD via changing the composition of gut microbiota has been reported in several studies; however, the results are not uniform [1,5,72]. Antibiotics can exacerbate existing conditions and may also contribute to the occurrence of autoimmune diseases. In a case-control study conducted in the UK, the authors reported that early childhood exposure to antibiotics increases the risk of Crohn’s disease (CD) [5]. Their observation was in agreement with the hygiene hypothesis, where reduced exposure to infectious bacteria due to improved sanitary practices and antibiotics during early life was attributed to the rise in the incidence of allergy and asthma in developed Western countries [3]).

Overproduction of Th2 cells and IgE causes asthma; however, as discussed above, gut microbiota affects the Th2 subset and may be a potential treatment option for such types of asthma. This was reported by Mazmanian et al. [3], who demonstrated that GF mice have a higher Th2:Th1 ratio than CONV-R mice. However, the condition was improved following colonization of the mice by microbiota that contained *Bacteroides fragilis*. The use of microbiota as a treatment for overproduced Th2 cells can become a therapeutic option for patients with low quality of life due to severe asthma.

### 2.3. Fecal Material Transplantation for the Treatment of Autoimmune Diseases and Antimicrobial Resistant Infections

FMT is an emerging treatment used to re-establish the microbial diversity associated with normal gut microbiota and pathogen colonization resistance in a person with a dysbiosis-associated disease. It involves the transfer of fecal material from a healthy person to a sick person [73,74]. The fecal material can be transferred fresh or frozen with 200 to 300 g of feces suspended in 500 mL saline solution and administered to the recipient via nasogastric or nasoduodenal tube, colonoscopy, enema, or capsule [75,76]. The transfer of stool from a healthy person to a sick person dates back to the 17th century in veterinary medicine [75]. However, there is evidence that dates this practice back to 4th century China when human fecal suspensions were given orally to patients who had food poisoning or severe diarrhea [77,78]. In modern times (1958), the first use of fecal enema was for the treatment of pseudomembranous enterocolitis associated with *Micrococcus pyogenes* infections [79].

FMT is similar to probiotics, which have been widely used to ease gastrointestinal disorders. However, the microbial composition of probiotics is simple, and their constituent exogenous microbes may not be able to colonize the gut persistently compared to the consortium delivered by FMT. The microbes in probiotic products and the metabolites (e.g., SCFAs) that they produce can ameliorate symptoms and modify microbiome compositions during their transit through the gastrointestinal tract [80,81]. In contrast, FMT results in persistent colonization of the gut establishing a new microbiota.

FMT has come to the medical spotlight after its success was witnessed in the treatment of recurring *C. difficile* infections at the beginning of this century. Currently, numerous research studies are underway to use FMT for the treatment of several diseases, including immune-related health problems, such as IBD, multiple sclerosis, and IBS [11,74,77].

#### 2.3.1. *Clostridioides difficile* Infections

The modern use of FMT began in the 1950s in humans for the treatment of pseudomembranous colitis caused by *Micrococcus pyogenes (Staphylococcus)* [79]. However, a clinical cure was reported in 1970, when a patient with post-operative and post-antibiotic colitis, a possible *C. difficile* infection, recovered after FMT in a research setting [82]. *C**lostridioides difficile* is an opportunistic intestinal pathogen causing antibiotic-associated diarrhea and colitis, and complicating IBD by triggering or worsening the inflammatory flare [83]. Disruption of the normal balance of colonic microbiota as a consequence of antibiotic use or other stressors is believed to augment the pathogenesis of *C. difficile* infection [78,84].

Patients with recurrent *C. difficile* infection have a marked reduction in microbial diversity as evidenced by marked changes in the composition of Bacteroidetes and Firmicutes in their stool compared with patients who have had just one episode of *C. difficile* infection and healthy control subjects [85]. It has been suggested that restoration of normal gut microbiota via FMT therapy could be beneficial as an alternative therapeutic approach to treat *C. difficile* infections. For instance, 97 patients with chronic *C. difficile* infections received FMT at the Mayo Specialized Clinic, where 85 (87.6%) of them recovered after the first FMT, five more recovered after the second FMT, while seven of them resorted to antibiotic use due to FMT failure [86]. FMT was also effective in treating nonrecurring or acute *C. difficile* infections [82]. In 2013, FMT was approved by the U.S. Food and Drug Administration for the treatment of recurring diarrhea caused by antibiotic-resistant *C. difficile* [9].

#### 2.3.2. Inflammatory Bowel Disease

IBD is a recurring chronic inflammation of the intestine, which consists of two main conditions: Crohn’s disease (CD) and ulcerative colitis (UC) [87]. Current standard therapies for IBD involve oral immunosuppressive agents; however, relapse and severe adverse effects remain problems [88]. Recent research evidence has indicated that perturbation of gut microbiota is involved in the pathogenesis of IBD. An inappropriate immunologic response to intestinal bacteria and a disruption in the balance of the gastrointestinal microbiota in genetically susceptible individuals drive the pathogenesis of IBD [89,90,91]. Numerous studies have also demonstrated a marked decline in intestinal microbial diversity and reduction in dominant intestinal bacteria or anti-inflammatory bacteria such as Firmicutes, *Lactobacillus*, and *Bacteroides*. In contrast, harmful bacteria, such as *Enterococcus* and Proteobacteria (e.g., adherent invasive *E. coli* (AIEC) strain), have increased in IBD patients [92,93].

In a rodent model, correlations between IBD and a decrease in Firmicutes and Bacteroidetes, and an increase in Proteobacteria and Actinobacteria were reported [94]. However, a systematic review in which the microbiota of IBD patients and control subjects was compared has revealed a lack of consistency between studies [95]. On the other hand, research has suggested that the aberrant gut microbial milieu created in IBD patients favors colonization and infection of the intestine easily by some opportunistic pathogens, especially *C. difficile* [83] and adherent and invasive *E. coli* (AIEC) [96]. Altogether, it can be concluded that the disruption of gut microbiota has an association with IBD, and thus re-establishment of the microbiota is helpful. While there is variability among studies as to the outcome of FMT therapies in IBD patients, several human and animal studies have reported promising results in which healthy gut microbiota has been restored, and UC symptoms have been lessened following FMT. FMT has been shown to have more benefit to UC than CD patients [97,98]. A few human and animal studies are summarized below.

In a mouse model, FMT has been found to ameliorate experimental UC by regulating gut microbiota, which might suggest the suitability of FMT for the control of UC [99]. The authors reported: (1) reduction in the level of disease activity index and improved body weight, colon weight, and length of time for which the mice remained in the study; (2) lessened histopathological changes, and decreased expression of key cytokines (e.g., TNFα, IFNγ, IL-7) and oxidative status in the colon; (3) downregulated expression of genes associated with the NF-kappaB signaling pathway; (4) restored gut microbiota to the pattern similar to the control group by enriching the relative abundance of Firmicutes and lowering the abundances of Bacteroidetes and Proteobacteria, and increasing the relative abundances of *Lactobacillus*, *Butyricicoccus*, *Lachnoclostridium*, *Olsenella*, and *Odoribacter* but decreasing *Helicobacter*, *Bacteroides,* and *Clostridium* species; and (5) increased the production of SCFAs in the colon of the experimental mice. The success has been attributed to the reduction in pro-inflammatory substances such as TNF-α and IL-1β, and an increased level of IL-10 in colonic tissues of experimental mice [87,100,101].

Similarly, a positive response has been reported in CD patients after receiving FMT. In one study, the clinical remission was two-fold higher in the patients who underwent FMT than in those in the control group [102]. In contrast, one study reported a lack of apparent beneficial effect of FMT on the clinical course of IBD even though it was effective in treating recurring *C. difficile* infections in IBD patients [103]. The inconsistency among literature reports may suggest that more work needs to be done to optimize the clinical application of FMT for IBD, given that CD and UC have a spectrum of underlying etiologies (i.e., adverse immune responses to microbial antigens). At the same time, the effectiveness of FMT in alleviating IBD symptoms in some cases holds promise for its broader use in the treatment of IBD in the future.

#### 2.3.3. Multiple Sclerosis

Multiple sclerosis (MS) is an inflammatory demyelinating disease of the central nervous system (CNS) with manifestations of varied neurological and autoimmune symptoms [104]. This chronic degenerative disease of unknown definitive cause and without cure increases by 200 cases every week, and there is an estimated total number of 400,000 cases in the U.S. according to epidemiological evidence [104]. Recently, data from several studies suggested that the interactions between the immune system and gut microbiota play a significant role in the pathogenesis of MS [73,105]). The mechanisms by which gut microbiota affect the pathogenesis of MS are yet to be unraveled; however, a study shows that MS patients have an altered microbiome, increased intestinal permeability, and changes in bile acid metabolism [106]. Moreover, it has been discovered that the brain–gut axis is bidirectional communication between CNS and the gut. Besides the CNS affecting the function of the gastrointestinal tract, gut microbiota/their metabolic products modulate the CNS [107,108]. An altered Schaedler flora mouse model, mice colonized by only eight bacterial species, has been recently demonstrated to serve as a unique model to elucidate mechanisms governing the microbiota–gut–brain axis [109].

Alterations of specific taxa in MS patients are associated with the promotion of inflammatory cytokines and overall inflammation together with a functional deficit of T regulatory cells [110,111,112]. Furthermore, there is experimental evidence that shows microbiota controls the levels of serotonin (5-hydroxytryptamine) in the gut via secretion and through regulation of metabolites. Conversely, alterations of gut microbiota affect the expression of 5-hydroxytryptamine transporters that are associated with MS [105]. FMT has an anti-inflammatory effect in MS cases, which is presumably associated with its promotion of IL-10 and TGF-β (transforming growth factor-beta) production and aryl hydrocarbon receptor upregulation [113].

The abovementioned human studies have been further corroborated by experimental autoimmune encephalomyelitis (EAE) as an animal model of MS [111]. Certain types of gut microbiota that have pro-inflammatory effects on the pathogenesis of MS, and other commensal bacteria and their antigenic products that protect against inflammation within CNS have been reported in mouse studies [111,114]. Specific bacterial taxa elevated or decreased in MS patients compared to healthy control were reviewed by Schepici and colleagues [73]. It is possible that the FMT introduces bacteria noted as reduced or missing in MS patients, such as *Parabacteroides* and *Adlercreutzia* [115].

In agreement with the mouse model, secondary-progressive MS was successfully treated with FMT in a case reported from Canada [116]. In the case study, a woman who had been living with secondary-progressive MS developed several bouts of *C. difficile* enterocolitis following clindamycin treatment of a gingival infection. Developing resistance to metronidazole and vancomycin had led her to receive a single FMT from her partner. Fortunately, the therapy halted the progression of neurological symptoms and stabilized secondary-progressive MS over the next ten years [116]. Several mouse models and case studies have also shown that gut microbiota modulation is a promising intervention approach for the management of MS [107]. Given that MS is a complex autoimmune disease with clinical variability, FMT can be used as an individualized therapeutic intervention rather than a “one size fits all” approach.

#### 2.3.4. Other Autoimmune Diseases

Epidemiological trends show that the numbers of allergy and asthma cases have increased globally in the last 50 years, which falls in line with the hygiene theory. This theory hypothesizes that the modern urban lifestyle reduces the exposure of children to naturally occurring microbes, which in early life may have prompted early immune maturation and prevented allergic diseases and asthma from developing [117]. Children growing up in rural areas, around animals, and in larger families seem to develop allergies and asthma less often than do other children [118]. Taken together, the modern lifestyles that are featured by high antibiotic consumption and less exposure to diverse environmental microbes during early childhood are believed to contribute to the rising incidence of asthma and allergies. In agreement with the hygiene hypothesis, human and animal studies suggest that regulation of immune response associated with the development of asthma is affected by gut microbiota [119]. There are research attempts to use FMT to treat asthma; however, considerable efforts are yet required to find an effective approach [119].

IBS is a common gastrointestinal disorder characterized by chronic and recurrent symptoms such as constipation, diarrhea, bloating, and abdominal pain without detectable biochemical or structural abnormalities [120]. Evidence from several studies has indicated an association between IBS and disturbance of microbial compositions and functions; interestingly, FMT has been shown to improve IBS symptoms [121,122,123]. Recently, three trials have been conducted to evaluate IBS responses to FMT therapies, where two of them have been successful, and an 89.1% response level has been recorded in one of the trials [124].

Furthermore, recent evidence from the literature suggests that gut microbiota can influence ocular health and inflammation, the existence of a gut–eye–lacrimal gland axis, suggesting that FMT might be considered as a therapeutic alternative [125]. FMT has also been shown to improve inflammation in necrotizing enterocolitis [126], neuroinflammation in murine cirrhosis [127], insulin resistance and repairing impaired islets in type 2 diabetes [128], and other conditions [77,129].

### 2.4. Challenges and Prospective of FMT

Several challenges limit the broader use of FMT. Developing appropriate procedures for the selection of donors and discovering new techniques that will allow for precise examination of donors’ fecal material in terms of the content of harmful metabolites and pathogenic microorganisms is a big challenge [130]. Recruitments of donors are stringent and involve screening for health and some other factors that can affect the gut microbiota. The donor has to be checked for the presence of human immunodeficiency virus (HIV); syphilis; hepatitis A, B, and C; autoimmune and atopic diseases; and other conditions such as tumors, inflammation, diabetes, various infectious diseases, and metabolic syndromes. The donor should not be overweight, drink alcohol, or use narcotics or drugs that could affect intestinal microbiota, such as immunosuppressants, steroids, probiotics, aspirin, proton pump inhibitors, antibiotics, etc. [130]. The gender of the donor also needs to be taken into consideration during the transplantation, as gender differences may contribute to microbial variation between males and females [131].

Furthermore, FMT entails some risks to the recipients, which include the acquisition of antimicrobial-resistant bacteria or resistance genes, inflammation, exacerbation of autoimmune diseases, and sepsis [9,82,84,102]. Recently, extended-spectrum beta-lactamase-producing *E. coli* bacteremia occurred in two patients who received FMT materials from the same stool donor, of which one was fatal [132]. Strikingly, a recent mouse model study shows FMT introduces segmented filamentous bacteria that may have a negative effect on bone development, which is related to the enhancement of Th17 responses [133]. Mechanical damage of tissues or intestinal perforation can happen during fecal transplantation from inappropriate endoscopic techniques [134]. Additionally, preplanning and preparing for FMT is challenging since long-term stability and storage of donated fecal materials can cause a reduction in the viability of the bacteria [82]. The latter challenge, however, can be overcome by opening fecal material donation and storage banks, which have recently been started in some places in the U.S. The storage of fecal microbiota requires anaerobic conditions.

In the future, transplantation of a selective bacterial consortia can be considered to replace ingestion or infusion of crude fecal material for many reasons, including its aesthetic value and ability to scale-up the availability of safe and reliable materials [78,135]. This practice has been coined as selective microbiota transplantation (SMT) [77]. Many human and animal studies have demonstrated that the effectiveness of SMT is comparable to that of FMT [136,137,138]. However, SMT requires examining the bacterial taxa that need to be administered at least for every condition, if not for every case, and filtering out of the required bacteria from the bulky fecal microbial community. This may not always be possible, as many of these bacteria cannot be cultured with current knowledge.

### 2.5. Association of Gut Microbiota with Immune Diseases and Fecal Material Transplantation in Animals

Similar to humans, the digestive tracts of animals are colonized by myriads of microbes, which play a significant role in the development of the immune system [139]. The effects of alterations of gut microbiota on the immunity and immune-associated disease conditions have been reported in cattle [140], chickens [141,142], horses [143,144], pigs [69,81,145], and dogs and cats [18,146]. For instance, in an experiment that was conducted on day-old chickens, it was shown that chickens with gut microbiota depleted by antibiotics had significantly downregulated type I interferon responses to influenza virus compared to nondepleted chickens [141]. Another study conducted in broiler chickens also indicated that perturbation of gut microbiota by early-life antibiotic treatment affected the expression of mucosal genes associated with intestinal immune development [147]. On the other hand, restoration of healthy gut microbiota by transferring fecal materials from healthy animals to sick animals can lead to clinical recovery for disease conditions associated with alteration of gut microbiota.

FMT, referred to as transfaunation using rumen fluid, has been used in veterinary medicine, particularly in ruminants, for a long time before gut microbiota was studied [14,148]. Several research trials have been performed to use FMT for the treatment of recurring diarrhea, IBD, mastitis, colitis, and other problems in animals [19,140,145,148,149,150]. Besides the transfer of fecal material from healthy to sick animals as performed in human medicine, FMT is also performed to increase the resistance of young animals to pathogens by transferring fecal material from adult animals to young animals. Fecal material can also be transferred from a resistance breed to a breed that is susceptible to a specific pathogen [149,150,151].

There are two critical factors that make FMT therapy appealing in the livestock industry: (1) The rise in the incidence of AMR by pathogens of veterinary importance. The emergence of AMR has put much pressure on animal agriculture to resort to an alternative approach to treat intestinal disorders, reproductive problems, and mastitis. (2) Increasing preference of consumers for antibiotic-free animal products such as meats, eggs, and milk [152]. Organic farmers are looking for alternative approaches to the conventional use of antibiotics to prevent and control disease, which would allow them to satisfy the consumer demands of the 21st century. Along this line, several research trials have been performed to curtail the shedding of foodborne pathogens, such as *Campylobacter jejuni* in poultry and other food animals, by transferring fecal materials from adult to young or from a resistance breed to a susceptible breed [19,153]. In the following section, a review of FMT therapy in pigs, dogs, and cats, owing to their potential use as a model to study human autoimmune diseases and FMT procedures, is presented.

#### 2.5.1. Pigs

A study was conducted in newborn piglets to evaluate the effects of early-life environmental variations on intestinal microbiota and immune development [69]. In the study, piglets in one group were kept undisturbed under a natural condition. Whereas the second and the third groups received a single dose of tulathromycin injection on day four after birth, and the latter group was subjected to stressful management procedures (i.e., docking, clipping, and weighing) as well. The three main findings after euthanizing the piglets on day eight were: (1) the diversity of microbiota increased in the treatment piglets compared to the control, which was considered by the authors as “more chaotic” (i.e., higher variability); (2) the relative abundance of beneficial bacteria, such as *Lactobacilli*, which are thought to have immunomodulatory effects, was significantly reduced in the treatment groups; (3) expression of toll-like receptor (TLR) and chemokine signaling networks that play a significant role in intestinal immunity were lower in the treatment cohorts than the control cohort. The authors suggested that disturbance of the gut microbial diversity in early-life, during the period in which the functions and structures of immunity are evolving, affects immune development [69]. Similarly, dietary supplementation with *Lactobacilli* to counteract the reduction in the abundance of intestinal *Lactobacilli* around weaning evoked immunomodulatory properties in the piglet intestine. It led to the upregulation of the expression of IL-4 and IFN-α in the cecum and downregulation of anti-inflammatory cytokine TGF-β1 in the small intestine [154].

FMT therapy has been attempted to control disease conditions in pigs [20,145,151]. Additionally, a standardized FMT preparation protocol for pigs has been outlined by Hu and colleagues [81]. In one study, after newborn piglets received fecal material from adult pigs: (1) relative abundance of harmful bacteria such as *Escherichia* decreased while the abundance of beneficial bacteria such as Firmicutes increased; (2) incidences of diarrhea decreased; (3) expressions of antimicrobial peptides, TLR2 and TLR4, and secretory IgA contributing to gut health increased [145]. The study demonstrated that FMT therapy played a beneficial role in the maturation of mucosal immunity (e.g., increased production of secretory IgA) via modulating the composition of the gut microbiota in the recipient piglets.

Another study has elucidated that the success of FMT is greatly influenced by the microbial compositions of donors, which exhibit significant variations between swine breeds. Three-day-old suckling piglets (*n* = 24) received intact fecal microbiota from Tibetan, Yorkshire, or Rongchang pigs [21]. However, only those piglets that received feces from the Tibetan pig showed improved gut development measured in terms of increased count of *Lactobacilli* species, reduced diarrhea index, and increased abundance of IL-10 mRNA in the colon compared to the control piglets and those who received FMT from other breeds. So far, prior studies have limitations, and thus more studies are needed to understand the role of breed differences in gut microbial diversity and composition in order to optimize the FMT donor selection approach in pigs.

On the other hand, piglets can be used to study the effects of microbiota on human health as an alternative to mice due to a closer genetic similarity between pigs and humans. The relationships between porcine immune system and human immune system have been shown in the study that compared the microbiota of rural Amish and non-Amish urban residents, where fecal materials collected from infants were transferred to GF piglets [155]. There were notable differences in the microbiome diversity and structure between the fecal samples of the infants from the two communities. Interestingly, the pig model has recapitulated the differences between the Amish and the non-Amish infants, which could be important to study how the microbiota affects the development of the mucosal immune system. Dhakal and colleagues [155] suggested further study to investigate additional advantages that the pig model offers over the mouse/rodent model.

#### 2.5.2. Dogs and Cats

The imbalance in gut microbiota has a significant impact on health, including the immune system in dogs and cats [156,157]. Similar to the case of human autoimmune diseases, acute and chronic gastrointestinal disorders, such as IBD, are associated with alterations of intestinal microbial communities in these companion animals [80,158,159,160]. In dogs, the association between gut microbial dysbiosis with IBD and inflammatory enteropathies has been well documented [161]. Similarly, FMT has been attempted for canine parvovirus infection [162], acute diarrhea [163], idiopathic IBD [17,164], and *C. difficile*-associated diarrhea [165]. The studies cited here reported successful results after FMT, although some studies lacked a control group and some were case reports; thus, future large-scale assessments are required.

As in the case of dogs, there have been successful FMT trials in cats. In a 10-year-old female cat with UC, FMT was attempted as a last therapeutic option before euthanasia. Fortunately, the authors reported an immediate response to the first therapy, and the clinical cure of diarrhea and passing of normal feces was achieved after 11 months of follow-up [18]. However, extensive studies are required to generate large data upon which evidence-based decisions can be made to use FMT in the treatment of gastrointestinal disorders in small animals. Nevertheless, veterinarians are encouraged to consider FMT in situations where other treatment options fail. Commentary on crucial aspects of FMT in small animal practice has been presented by Chaitman et al. [166].

Similar to pigs, small animals can be used as a model to study autoimmune diseases and FMT therapy in humans. Comparisons of microbial shifts between the human, dog, and cat IBD are summarized by Honneffer et al. [158]. Interestingly, several bacterial taxa associated with IBD are shared between humans and dogs, which may suggest that gut microbiota associated with a specific autoimmune disease might be conserved in several mammals.

## 3. Discussion

Gut microbiota is considered an additional organ for humans and animals because of its irreplaceable vital functions [167]. Given that, restoration of perturbed microbiota by using FMT may improve health in humans and productivity, welfare, and health in animals. FMT can be considered as a re-emerging therapy since it has been utilized for a long time but abandoned for decades due to the discovery of antimicrobials in the mid-20th century. The interest toward this re-emerging therapy has recently been revived due to the following three main reasons: (1) lack of effective and lasting treatment options for some diseases, such as recurrent *C. difficile* infections, IBD, and IBS; (2) the increased demand for alternative approaches to antibiotics to treat gastrointestinal bacterial infections due to a rise in antibiotic resistance; and (3) increased preference of consumers for antibiotic-free animal products, such as meats, eggs, and milk, a phenomenon that compels farmers to seek alternative methods to antibiotics to prevent and control diseases in food animals.

In summary, the literature provides robust evidence regarding the significant impacts that gut microbiota has on the development of the immune system and its association with several immune-related diseases in humans and animals. FMT has been effective in the treatment of recurrent *C. difficile* infections as well as shown to improve gut decolonization from other antibiotic-resistant bacterial species. Additionally, this therapy has provided promising outcomes in IBD, MS, and IBS cases, despite some irregularities among studies. Besides holding promise for the treatment of autoimmune diseases in small animals, FMT has high potential to promote colonization resistance toward pathogens in young or susceptible animals and to decrease shedding and spreading of foodborne pathogens on poultry and swine farms.

Further studies are needed: (1) to understand more about the mechanisms by which critical members of the gut microbiota modulates immune development and physiological responses that are associated with development of autoimmune diseases; (2) to improve methods of donor recruitments, fecal material collection, characterization, storage, preparation, scale-up, and administration to minimize the side effects of FMT on recipients and make it more appealing; and (3) to demonstrate that the use of FMT in veterinary medicine may prevent and control diarrheal diseases, respiratory diseases, mastitis, colitis, and shedding and spread of foodborne pathogens in farm animals using this (re)emerging approach.

## Figures and Tables

**Figure 1 antibiotics-11-01093-f001:**
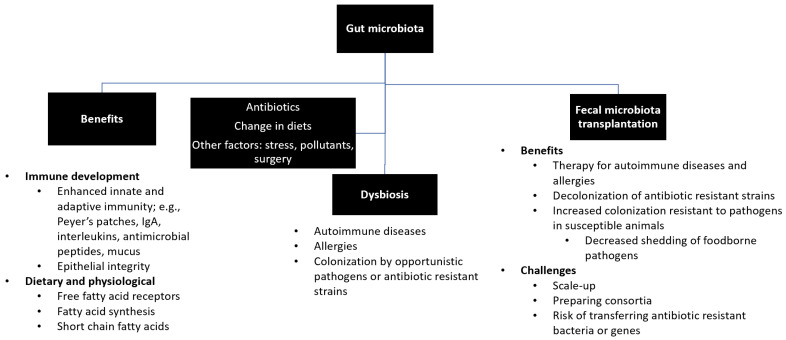
Summary of gut microbiota. Gut microbiota plays beneficial roles in the development of immune system, production and synthesis of essential nutrients, and maintenance of a healthy physiological status. Exposure to antibiotics and changes in diets and lifestyles may lead to disruption of healthy microbial diversity and composition (i.e., dysbiosis), which can be restored by fecal microbiota transfer.

## Data Availability

Not applicable.

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
