# Peer review of "Impacts of Gut Microbiota on the Immune System and Fecal Microbiota Transplantation as a Re-Emerging Therapy for Autoimmune Diseases"

_antibiotics, 2022, doi:10.3390/antibiotics11081093_

Round 1

Reviewer 1 Report

This manuscript reviews the influence of gut microbiota on the immune response and the association between gut microbiota and several major autoimmune diseases in humans. Then the authors summarize the use of FMT as an antibiotic alternative approach to these autoimmune diseases. This paper also includes research on FMT applied in disease conditions of animals (pig, dog, and cat). Overall, this review is well written, includes adequate information, and will be useful for researchers interested in FMT.   

Specific comments

 Title: Change to “Impacts of Gut Microbiota on the Immune System and Fecal Microbiota Transplantation as a Re-emerging Therapy for Autoimmune Diseases”

Line 25: change ‘the potential FMT has as the next generation…’ to ‘FMT has the potential as the next generation…’’

Lines 50-53: Plenty of research has demonstrated the association between gut microbiota and many diseases, including autoimmune diseases and other types of diseases. All “cancer, mental health, and cardiovascular, respiratory, and metabolic diseases” (as you mentioned in the manuscript) can be correlated with gut microbiota dysbiosis (doi: 10.1038/s41392-022-00974-4; doi: 10.1038/s41579-020-0433-9; doi: 10.1038/s41591-022-01779-2). Please confirm the role of gut microbiota in health and disease and rewrite this part.

Lines 110-111: Change “The perturbation of gut microbiota may lead to a decrease in microbial diversity and composition …” to “The perturbation of gut microbiota may lead to a decrease in microbial diversity and a shift of microbiota composition…”.

Lines 109-177, Section 2.1.2. Factors affecting diversities and compositions of gut microbiota: This review is focused on the effect of gut microbiota on immunity, autoimmune diseases, and FMT as the therapy for autoimmune diseases. It looks like the inclusion of the 2.1.2 section is not necessary and deleting this part is fine. Alternatively, summarize factors affecting gut microbiota in brief and discard lines 150-177.

Lines 547-549: It is much easier to control the environment and diet of mice than that of pigs. Mice are much cheaper and easier to be managed than pigs when the two species are used as animal models. Closer genetic similarity between humans and pigs than with mice is a major reason for pig as a more potentially suitable animal model to study the immunity/gut microbiota of humans. Correction of this sentence is needed.

When citing research studies, it is not necessary to describe the study (e.g., objectivity, experimental design, methods, etc.) in detail (e.g., lines 508-522). Besides, it would be redundant to describe studies one by one in a review when there are multiple pieces of literature to cite. A summary of the major findings linked with the review topic is better. It is strongly recommended to go through this manuscript and make it concise. 

Author Response

Date: August 08, 2022

MDPI journal Antibiotics

We are submitting the revised version of our review article entitled “Impacts of Gut Microbiota on the Immune System and Fecal Microbiota Transplantation as a Re-emerging Therapy for Autoimmune Diseases”. We are grateful to the editor and the reviewers for the valuable comments and suggestions. All the suggestions and comments have been addressed in the revised version of the manuscript. As suggested a figure that summarizes this review article has been also included in this revised version. Please, find the detailed responses below.

Reviewer # 1

This manuscript reviews the influence of gut microbiota on the immune response and the association between gut microbiota and several major autoimmune diseases in humans. Then the authors summarize the use of FMT as an antibiotic alternative approach to these autoimmune diseases. This paper also includes research on FMT applied in disease conditions of animals (pig, dog, and cat). Overall, this review is well written, includes adequate information, and will be useful for researchers interested in FMT.   

Thank you so much for your time to review this paper and providing us your thoughtful comments and suggestions; definitely, they are useful to improve the quality of this manuscript. Please, find our responses to your specific comments below.

Specific comments

 Title: Change to “Impacts of Gut Microbiota on the Immune System and Fecal Microbiota Transplantation as a Re-emerging Therapy for Autoimmune Diseases”

Modified according to the comments

Line 25: change ‘the potential FMT has as the next generation…’ to ‘FMT has the potential as the next generation…’’

Changed according to the comment.

Lines 50-53: Plenty of research has demonstrated the association between gut microbiota and many diseases, including autoimmune diseases and other types of diseases. All “cancer, mental health, and cardiovascular, respiratory, and metabolic diseases” (as you mentioned in the manuscript) can be correlated with gut microbiota dysbiosis (doi: 10.1038/s41392-022-00974-4; doi: 10.1038/s41579-020-0433-9; doi: 10.1038/s41591-022-01779-2). Please confirm the role of gut microbiota in health and disease and rewrite this part.

Thank you for the comment and the 3 review articles. The following statement is added for clarification and confirmation of the role of gut microbiota in disease. Moreover, figure 1 clarifies the health benefits and the association of gut microbiota with different disease conditions.

“Furthermore, disruption of gut microbial diversity and composition has been shown to lead to the occurrence of several health problems including autoimmune diseases”

Lines 110-111: Change “The perturbation of gut microbiota may lead to a decrease in microbial diversity and composition …” to “The perturbation of gut microbiota may lead to a decrease in microbial diversity and a shift of microbiota composition…”.

 Changed according to the comment.

Lines 109-177, Section 2.1.2. Factors affecting diversities and compositions of gut microbiota: This review is focused on the effect of gut microbiota on immunity, autoimmune diseases, and FMT as the therapy for autoimmune diseases. It looks like the inclusion of the 2.1.2 section is not necessary and deleting this part is fine. Alternatively, summarize factors affecting gut microbiota in brief and discard lines 150-177.

We appreciate your comments on this section. We made some changes to its contents; however, we did not take this section out because these factors are assumed to contribute to the occurrence of dysbiosis, which may ultimately lead to autoimmune diseases and other health conditions. The interrelationships between these factors (e.g., diets, antibiotics), dysbiosis, autoimmune diseases and FMT are presented in the figure included in the revised version of the manuscript.

Lines 547-549: It is much easier to control the environment and diet of mice than that of pigs. Mice are much cheaper and easier to be managed than pigs when the two species are used as animal models. Closer genetic similarity between humans and pigs than with mice is a major reason for pig as a more potentially suitable animal model to study the immunity/gut microbiota of humans. Correction of this sentence is needed.

Modified according to this comment.

When citing research studies, it is not necessary to describe the study (e.g., objectivity, experimental design, methods, etc.) in detail (e.g., lines 508-522). Besides, it would be redundant to describe studies one by one in a review when there are multiple pieces of literature to cite. A summary of the major findings linked with the review topic is better. It is strongly recommended to go through this manuscript and make it concise. 

We appreciate your comments and we have edited several sections based on your suggestion (e.g., lines 140-146).

Reviewer #2

The article presents a review on gut microbiota and the re-emerging potential of FMT (which has been used since ancient times without the proper knowledge). The information is presented clearly, concise and understandable and it would be of great use for researchers interested in these topics.

Thank you so much for your time to review this paper and providing us your thoughtful comments and suggestions; definitely, they are useful to improve the quality of this manuscript. Please, find our responses to your specific comments below.

 Elements that should be modified:  

There is an absence of images/graphics. Those would make the article exceptional. 

A figure that summarizes the main points in this review manuscript is provided in this revised version.

I would suggest a graphic or a picture representing human microbiota with some of the information gathered from lines 85 to 93 (Section 2.1).  

We appreciate your suggestions. We agree that a figure is better than a text to present proportions of phyla in huma gut microbiota. However, we think that this information is not complex, only the relative abundances of Firmicutes and Bacteroidetes are cited here.

 A minor change that would also be good would be to summarize Section 2.1.2. The elements presented in this section are there to point out a few important aspects but are not necessary.  

Thank you for your comment. We made some changes to this section; however, we did not take this section out because these factors are assumed to contribute to the occurrence of dysbiosis, which may ultimately lead to autoimmune diseases and other health conditions. The interrelationships between these factors (e.g., diets, antibiotics), dysbiosis, autoimmune diseases and FMT are presented in the figure included in the revised version of the manuscript.

Line 619 on the next page.

This should be corrected in final the proofreading.

Reviewer 2 Report

The article presents a review on gut microbiota and the re-emerging potential of FMT (which has been used since ancient times without the proper knowledge). The information is presented clearly, concise and understandable and it would be of great use for researchers interested in these topics .   

Elements that should be modified:  

There is an absence of images/graphics. Those would make the article exceptional. 

I would suggest a graphic or a picture representing human microbiota with some of the information gathered from lines 85 to 93 (Section 2.1).  

 A minor change that would also be good would be to summarise Section 2.1.2. The elements presented in this section are there to point out a few important aspects but are not necessary.  

Line 619 on the next page.

Author Response

Date: August 08, 2022

MDPI journal Antibiotics

We are submitting the revised version of our review article entitled “Impacts of Gut Microbiota on the Immune System and Fecal Microbiota Transplantation as a Re-emerging Therapy for Autoimmune Diseases”. We are grateful to the editor and the reviewers for the valuable comments and suggestions. All the suggestions and comments have been addressed in the revised version of the manuscript. As suggested a figure that summarizes this review article has been also included in this revised version. Please, find the detailed responses below.

Reviewer # 1

This manuscript reviews the influence of gut microbiota on the immune response and the association between gut microbiota and several major autoimmune diseases in humans. Then the authors summarize the use of FMT as an antibiotic alternative approach to these autoimmune diseases. This paper also includes research on FMT applied in disease conditions of animals (pig, dog, and cat). Overall, this review is well written, includes adequate information, and will be useful for researchers interested in FMT.   

Thank you so much for your time to review this paper and providing us your thoughtful comments and suggestions; definitely, they are useful to improve the quality of this manuscript. Please, find our responses to your specific comments below.

Specific comments

 Title: Change to “Impacts of Gut Microbiota on the Immune System and Fecal Microbiota Transplantation as a Re-emerging Therapy for Autoimmune Diseases”

Modified according to the comments

Line 25: change ‘the potential FMT has as the next generation…’ to ‘FMT has the potential as the next generation…’’

Changed according to the comment.

Lines 50-53: Plenty of research has demonstrated the association between gut microbiota and many diseases, including autoimmune diseases and other types of diseases. All “cancer, mental health, and cardiovascular, respiratory, and metabolic diseases” (as you mentioned in the manuscript) can be correlated with gut microbiota dysbiosis (doi: 10.1038/s41392-022-00974-4; doi: 10.1038/s41579-020-0433-9; doi: 10.1038/s41591-022-01779-2). Please confirm the role of gut microbiota in health and disease and rewrite this part.

Thank you for the comment and the 3 review articles. The following statement is added for clarification and confirmation of the role of gut microbiota in disease. Moreover, figure 1 clarifies the health benefits and the association of gut microbiota with different disease conditions.

“Furthermore, disruption of gut microbial diversity and composition has been shown to lead to the occurrence of several health problems including autoimmune diseases”

Lines 110-111: Change “The perturbation of gut microbiota may lead to a decrease in microbial diversity and composition …” to “The perturbation of gut microbiota may lead to a decrease in microbial diversity and a shift of microbiota composition…”.

 Changed according to the comment.

Lines 109-177, Section 2.1.2. Factors affecting diversities and compositions of gut microbiota: This review is focused on the effect of gut microbiota on immunity, autoimmune diseases, and FMT as the therapy for autoimmune diseases. It looks like the inclusion of the 2.1.2 section is not necessary and deleting this part is fine. Alternatively, summarize factors affecting gut microbiota in brief and discard lines 150-177.

We appreciate your comments on this section. We made some changes to its contents; however, we did not take this section out because these factors are assumed to contribute to the occurrence of dysbiosis, which may ultimately lead to autoimmune diseases and other health conditions. The interrelationships between these factors (e.g., diets, antibiotics), dysbiosis, autoimmune diseases and FMT are presented in the figure included in the revised version of the manuscript.

Lines 547-549: It is much easier to control the environment and diet of mice than that of pigs. Mice are much cheaper and easier to be managed than pigs when the two species are used as animal models. Closer genetic similarity between humans and pigs than with mice is a major reason for pig as a more potentially suitable animal model to study the immunity/gut microbiota of humans. Correction of this sentence is needed.

Modified according to this comment.

When citing research studies, it is not necessary to describe the study (e.g., objectivity, experimental design, methods, etc.) in detail (e.g., lines 508-522). Besides, it would be redundant to describe studies one by one in a review when there are multiple pieces of literature to cite. A summary of the major findings linked with the review topic is better. It is strongly recommended to go through this manuscript and make it concise. 

We appreciate your comments and we have edited several sections based on your suggestion (e.g., lines 140-146).

Reviewer #2

The article presents a review on gut microbiota and the re-emerging potential of FMT (which has been used since ancient times without the proper knowledge). The information is presented clearly, concise and understandable and it would be of great use for researchers interested in these topics.

Thank you so much for your time to review this paper and providing us your thoughtful comments and suggestions; definitely, they are useful to improve the quality of this manuscript. Please, find our responses to your specific comments below.

 Elements that should be modified:  

There is an absence of images/graphics. Those would make the article exceptional. 

A figure that summarizes the main points in this review manuscript is provided in this revised version.

I would suggest a graphic or a picture representing human microbiota with some of the information gathered from lines 85 to 93 (Section 2.1).  

We appreciate your suggestions. We agree that a figure is better than a text to present proportions of phyla in huma gut microbiota. However, we think that this information is not complex, only the relative abundances of Firmicutes and Bacteroidetes are cited here.

 A minor change that would also be good would be to summarize Section 2.1.2. The elements presented in this section are there to point out a few important aspects but are not necessary.  

Thank you for your comment. We made some changes to this section; however, we did not take this section out because these factors are assumed to contribute to the occurrence of dysbiosis, which may ultimately lead to autoimmune diseases and other health conditions. The interrelationships between these factors (e.g., diets, antibiotics), dysbiosis, autoimmune diseases and FMT are presented in the figure included in the revised version of the manuscript.

Line 619 on the next page.

This should be corrected in the final proofreading.